# GATED RELATIONAL GRAPH ATTENTION NETWORKS

## ABSTRACT

Relational Graph Neural Networks (GNN), like all GNNs, suffer from a drop in performance when training deeper networks, which may be caused by vanishing gradients, over-parameterization, and over-smoothing. Previous works have investigated methods that improve the training of deeper GNNs, which include normalization techniques and various types of skip connection within a node. However, learning long-range patterns in multi-relational graphs using GNNs remains an under-explored topic. In this work, we propose a novel relation-aware GNN architecture based on the Graph Attention Network that uses gated skip connections to improve long-range modeling between nodes and uses a more scalable vector-based approach for parameterizing relations. We perform extensive experimental analysis on synthetic and real data, focusing explicitly on learning long-range patterns. The proposed method significantly outperforms several commonly used GNN variants when used in deeper configurations and stays competitive to existing architectures in a shallow setup.

## 1 INTRODUCTION

In this work, we focus on learning long-range patterns in multi-relational graphs using graph neural networks (GNN), which heavily relies on the ability to train deep networks[1]. However, GNNs suffer from decreasing performance when the number of layers is increased. Zhao & Akoglu (2020) point out that this may be due to (1) over-fitting, (2) vanishing gradients, and (3) over-smoothing (the phenomenon where node representations become less distinguishable from each other when more layers are used). Recently, several works have investigated over-fitting (Vashishth et al., 2020), over-smoothing (Li et al., 2018; Chen et al., 2019; Zhao & Akoglu, 2020; Rong et al., 2019; Yang et al., 2020), over-squashing (Alon & Yahav, 2020), and possible vanishing gradient (Hochreiter & Schmidhuber, 1997; Pascanu et al., 2013; He et al., 2016) problems in GNNs (Li et al., 2019a; 2020; Rahimi et al., 2018).

One simple but effective technique to improve the training of deeper GNNs is using skip connections, for example, implemented by the Gated Recurrent Unit (GRU) (Cho et al.) in the Gated GNN (GGNN) (Li et al., 2015). Such connections can improve learning deep GNNs as it avoids vanishing gradients towards lower-layer representations of the same node, and reduces over-smoothing (Hamilton, 2020). However, such vertical skip connections are not sufficient to enable learning long-range patterns. In addition, in relational GNNs, such as the Relational Graph Convolutional Network (RGCN) (Schlichtkrull et al., 2018), training difficulties may arise from the methods used for integrating relation information, which may suffer from over-parameterization and impact backpropagation.

In this work, we develop a novel GNN architecture for multi-relational graphs that reduces the vanishing gradient and over-parameterization problems that occur with existing methods and improves generalization when learning long-range patterns using deeper networks. Several changes are proposed to the Graph Attention Network (GAT) (Veličković et al., 2018), including a modified attention mechanism, an alternative GRU-based update function, and a gated relation-aware message function. An extensive experimental study is conducted that (1) shows that different existing relation-aware GNNs fail to learn simple patterns in a simple synthetic sequence-based graph classification task, (2) presents a comparison and ablation study on a synthetic node classification task, (3) shows that our architecture is competitive with existing ones on an entity classification task using real-world data from previous work.

---

[1]We need at least $K$ GNN layers to capture information that is $K$ hops away.

## 2 BACKGROUND: GNNs AS MESSAGE PASSING NETWORKS

Many popular GNNs can be formulated within a message passing (MP) framework (Gilmer et al., 2017). Consider a multi-relational graph $\mathcal{G} = (\mathcal{V}, \mathcal{E})$, where $\mathcal{E}$ consists of edges $e$ which are specified by a triple $(u, v, r)$ that define a directed edge from $u$ to $v$, labeled by edge type (relation) $r$. A GNN maps each node $v \in \mathcal{V}$ onto representation vectors $\mathbf{h}_v^1 \dots \mathbf{h}_v^K$ by repeatedly aggregating the representations of the immediate neighbours of every node and updating node representations in every step of the encoding process, each associated with one of $K$ layers of the GNN. Relational GNNs must also take into account the edge types between nodes in the graph $\mathcal{G}$. In the following sections, we use the message passing framework where a single GNN layer/step is decomposed into a three-step process:

$$\mathbf{h}_v^{(k)} = \phi(\mathbf{h}_v^{(k-1)}, \gamma(\{\mu(\mathbf{h}_u^{(k-1)}, r)\}_{(u,v,r) \in \mathcal{E}(\cdot, v)})) \ , \tag{1}$$

where $\mu(\cdot)$ computes a "message" along a graph edge using the neighbour representation $\mathbf{h}_v^{(k-1)}$ and the edge type $r$ of an edge from $u$ to $v$, $\gamma(\cdot)$ aggregates the incoming messages into a single vector, and $\phi(\cdot)$ computes a new representation for node $v$. $\mathcal{E}(\cdot, v)$ denotes the set of all edges in $\mathcal{G}$ that end in $v$. After subsequently applying Eq. 1 $K$ times to each node of the graph, the final node representations $\mathbf{h}_v^{(K)}$ can be used for different tasks, such as graph or node classification. One commonly used message function is a relation-specific linear transformation, which is used by RGCNs and GGNNs: $\mu_{\mathrm{MM}}(\mathbf{h}_u, r) = \mathbf{W}_r \mathbf{h}_u$, where $\mathbf{W}_r$ is a $\mathbb{R}^{D \times D}$ parameter matrix associated with edge type $r$. Moreover, GGNNs implement the update function $\phi(\cdot)$ using a GRU: $\mathbf{h}_v^{(k)} = \mathrm{GRU}(\overline{\mathbf{h}}_v^{(k-1)}, \mathbf{h}_v^{(k-1)})$, where the GRU's *input* argument is $\overline{\mathbf{h}}_v^{(k-1)} = \gamma(\{\mu(\mathbf{h}_u^{(k-1)}, r)\}_{(u,v,r) \in \mathcal{E}(\cdot, v)})$, the vector for the aggregated neighbourhood of $v$.

## 3 MOTIVATION: BREADTH-WISE BACKPROPAGATION

Many methods have very recently been proposed to mitigate vanishing gradients in GNNs, using techniques such as residual connections (Li et al., 2019a; 2020), "jumping knowledge" connections (Xu et al., 2018), and DenseNets (Huang et al., 2017), which have effects similar to the Gated GNN from Li et al. (2015). However, these techniques address only backpropagation in the depth of the network (i.e., *vertically* towards lower-level features of the same node). As we will now discuss, this is not sufficient for learning long-range patterns since breadth-wise backpropagation (i.e., *horizontally* towards the neighbours) in such GNNs may still result in learning problems and vanishing gradients.

Consider a simple graph convolutional network (GCN) with the general update of the hidden units $\mathbf{h}_v^{(k)} = \sigma(\mathbf{W}^{(\mathbf{k})}(\mathbf{h}_v^{(k-1)} + \sum_{(u,v) \in \mathcal{E}} \mathbf{h}_u^{(k)}))$, where $\sigma$ is a non-linearity such as a ReLU. Following the classical arguments, functions of this form will suffer from vanishing or exploding gradients because of the stacking of linear transformations ($\mathbf{W}$) and non-linearities ($\sigma$). The backpropagation path from the top-level features $\mathbf{h}_v^{(K)}$ of a node to its initial features $\mathbf{h}_v^{(0)}$ is $\sigma(\mathbf{W}^{(K)}(\sigma(\mathbf{W}^{(K-1)} \dots (\sigma(\mathbf{W}^{(1)} \mathbf{h}_v^{(0)}) \dots )))$. With many layers (i.e., when $K$ is large), the gradient magnitude diminishes at every step because of the multiplication with the derivative of the activation function, which is always $< 1$ for some choices. Additionally, depending on the values of the weights $\mathbf{W}$, repeated multiplication with the weights may cause the gradients to either vanish or explode. This could become especially problematic when sharing weights between layers, which could be useful to combat over-parameterization in deeper networks, which are necessary to encode multiple hops. These problems can be mitigated by the choice of activation function (such as using a ReLU), using good initializations for the weights, and using normalization layers. LSTMs (Hochreiter & Schmidhuber, 1997), GRUs (Cho et al.), ResNets (He et al., 2016), Highway Networks (Srivastava et al., 2015), and DenseNets (Huang et al., 2017) propose an alternative solution by introducing some form of skip connection around one or more layers, such that every layer contributes an update additively. This ensures that the gradient to deeper layers does not vanish, and also that the gradient w.r.t. features (or states) in different layers (or timesteps) is similar (GRUs, Highway Network) or the same (ResNets). In fact, He et al. (2016) point out that the improvement they obtain is not due to gradient magnitudes and attribute the effect to the fact that modeling the residual error is an easier task for the network.

Depth-wise backpropagation for the simple GCN can be improved by adapting it to a residual GCN (ResGCN) (Li et al., 2019a), such that we get the following update of the hidden units describing the node features: $\mathbf{h}_v^{(k)} = \mathbf{h}_v^{(k-1)} + \sigma(\mathbf{W}^{(\mathbf{k})}(\mathbf{h}_v^{(k-1)} + \sum_{(u,v)\in\mathcal{E}} \mathbf{h}_u^{(k)}))$. Alternatively, GGNN uses a GRU-based update function that implements a gated skip connection.

However, these techniques do not address breadth-wise backpropagation (towards the neighbouring nodes), which is essential for learning long-range patterns. For an illustration of this problem, consider the hidden update of the ResGCN. We can easily see that every backpropagation path from the top-level features $\mathbf{h}_v^{(K)}$ of a node $v$ to the $(K-L)$-level features $\mathbf{h}_u^{(K-L)}$ of a node $u$ that is $L$ hops away is $\sigma(\mathbf{W}^{(K)}(\sigma(\mathbf{W}^{(K-1)}\ldots(\sigma(\mathbf{W}^{(1)}\mathbf{h}_u^{(K-L)})\ldots)))$, which contains $L$ linear transformations and non-linearities. And all the backpropagation paths from $\mathbf{h}_v^{(K)}$ to $\mathbf{h}_u^{(0)}$ also contain $L$ such applications. A similar argument can be made for the GGNN, for which we provide a derivation of gradients in Appendix A.4.

While proper initialization, the use of normalization layers, and non-saturating nonlinearities may maintain good gradient magnitudes, the training of the network can still be improved by implementing some form of breadth-wise skip connections. In addition, as we will experimentally confirm, the use of *only depth-wise* skip connections can actually *hurt* performance when the task requires learning patterns containing distant nodes.

Note that the breadth-wise gradient issues are further exacerbated when averaging aggregators are used, which diminish the gradient towards a neighbour by a factor equal to the number of incoming edges. This leads to a gradient magnitude decay that is exponential in the distance between nodes. This drop is also related to the issues of oversmoothing and over-squashing (Alon & Yahav, 2020).

## 4 GATED RELATIONAL GRAPH ATTENTION NETWORKS

In this section, we describe a novel class of GNN architectures that addresses the breadth-wise backpropagation issues and prevents exponential breadth-wise gradient decay. We propose several changes which all follow the general principle of using additive (gated) feature updates not only depth-wise (vertically) but also breadth-wise (horizontally).

### 4.1 SYMMETRICALLY GATED RECURRENT UNIT

As elaborated above, depth-wise residual update functions and GRUs as used in the GGNN improve depth-wise backpropagation towards lower-level features of the same node. To also improve the breadth-wise backpropagation, we propose to use an adapted version of the GRU. We call the proposed update function, which is described by the following equations, a *Symmetrically Gated Recurrent Unit* (SGRU):

$$\mathbf{r}_h^{(k)} = \sigma(W_{r_h}\overline{\mathbf{h}}_v^{(k-1)} + U_{r_h}\mathbf{h}_v^{(k-1)} + b_{r_h}) \ , \quad \mathbf{r}_x^{(k)} = \sigma(W_{r_x}\overline{\mathbf{h}}_v^{(k-1)} + U_{r_x}\mathbf{h}_v^{(k-1)} + b_{r_x}) \quad (2)$$

$$\mathbf{z}_x^{(k)} = W_{z_x}\overline{\mathbf{h}}_v^{(k-1)} + U_{z_x}\mathbf{h}_v^{(k-1)} + b_{z_x} \ , \quad \mathbf{z}_h^{(k)} = W_{z_h}\overline{\mathbf{h}}_v^{(k-1)} + U_{z_h}\mathbf{h}_v^{(k-1)} + b_{z_h} \quad (3)$$

$$\mathbf{z}_u^{(k)} = W_{z_u}\overline{\mathbf{h}}_v^{(k-1)} + U_{z_u}\mathbf{h}_v^{(k-1)} + b_{z_u} \ , \quad \hat{z}_{x,i}^{(k)}, \hat{z}_{h,i}^{(k)}, \hat{z}_{u,i}^{(k)} = \text{softmax}([z_{x,i}^{(k)}, z_{h,i}^{(k)}, z_{u,i}^{(k)}]) \quad (4)$$

$$\hat{\mathbf{h}}_v^{(k)} = \tanh(W(\overline{\mathbf{h}}_v^{(k-1)} \odot \mathbf{r}_x^{(k)}) + U(\mathbf{h}_v^{(k-1)} \odot \mathbf{r}_h^{(k)})) \quad (5)$$

$$\mathbf{h}_v^{(k)} = \hat{\mathbf{z}}_x^{(k)} \odot \overline{\mathbf{h}}_v^{(k-1)} + \hat{\mathbf{z}}_h^{(k)} \odot \mathbf{h}_v^{(k-1)} + \hat{\mathbf{z}}_u^{(k)} \odot \hat{\mathbf{h}}_v^{(k)} \ . \quad (6)$$

The SGRU differs from the GRU used in the GGNN in (i) introducing an additional reset gate $\mathbf{r}_x^{(k)}$ that is applied to the aggregated neighbour states $\overline{\mathbf{h}}_v^{(k-1)}$ and (ii) computing the output state $\mathbf{h}_v^{(k)}$ as a three-way mixture between the previous node state $\mathbf{h}_v^{(k-1)}$, the aggregated neighbour states $\overline{\mathbf{h}}_v^{(k-1)}$, and the candidate state $\hat{\mathbf{h}}_v^{(k)}$ instead of a two-way mixture between $\mathbf{h}_v^{(k-1)}$ and $\hat{\mathbf{h}}_v^{(k)}$. The interpolation coefficients $\hat{\mathbf{z}}$ are produced by an elementwise softmax between the three $\mathbf{z}$ vectors. See also Figure 2 for an illustration.

The vanilla GRU as applied to sequences has two inputs: the input vector $x_t$ at a certain timestep $t$ and the state vector $h$. While it implements additive gated updates on $h$, it applies linear transformations

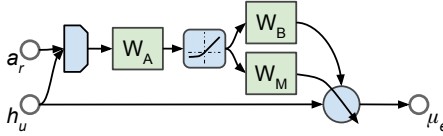

Figure 1: The proposed gated message function.

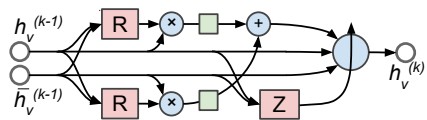

Figure 2: The proposed SGRU update function.

and nonlinearities on $x_t$ before merging it into $h$. In the SGRU, both inputs are gated similarly, which lets the $x_t$ input benefit from the same additive gradient behavior as $h$ and gave rise to the name "symmetrically gated". This is crucial for improving horizontal information flow during the training of GNN since the $x_t$ input of the SGRU receives the aggregated neighbourhood vector.

In synthetic task experiments (see Sections 5.1 and 5.2), we show that the SGRU is superior to the GRU as a GNN update function. However, this modification alone proved insufficient. To actually enable learning of long-range patterns, we also need to address the problem of exponential breadth-wise gradient decay and over-squashing (Alon & Yahav, 2020). This is addressed by the modifications described in the following sections.

## 4.2 ATTENTION-BASED NEIGHBOURHOOD AGGREGATION

As elaborated at the end of Section 3, averaging aggregators, which are commonly used to stabilize GNN training, lead to decaying gradient magnitudes w.r.t. distant nodes. In addition, a simple, non-weighted sum based aggregator squashes the entire $K$-hop *receptive field* of a $K$-layer GNN into a single vector uniformly, making it very challenging to distinguish features from the exponentially growing number of connected distant nodes (Alon & Yahav, 2020). For this reason, we believe that using an attention mechanism is crucial to enable the learning of long-range patterns in graphs. In the initial stages of training, when the attention weights are more uniform, the network might still suffer from exponentially decaying gradients. However, this problem should be reduced during training if attention learns to focus on a few specific node.

To implement the aggregation function $\gamma(\cdot)$, we adapt the scaled multi-head multiplicative attention mechanism (Vaswani et al., 2017). The per-head attention distributions are computed as described for transformers, with two small but important differences: (1) the edge type embedding are used directly in the keys when computing the attention, and (2) the linear transformations normally used to obtain value vectors are omitted. This last change follows the principle of using only additive operations to enable better long-range backpropagation through messages and node features. See Appendix A.1 for a more detailed description.

## 4.3 USING EDGE FEATURES

Different ways of using edge features have been proposed. RGCN and GGNN use multiplication with a relation-specific matrix ($\mu_{\text{MM}}$). CompGCN (Vashishth et al., 2020) and VR-GCN (Ye et al., 2019) make use of vector-based relation parameterization, and explore composition functions inspired by graph embedding learning methods (Bordes et al., 2013; Yang et al., 2014; Nickel et al., 2016). However, all these methods implement a linear transformation based update of the neighbour state.

To preserve the gradient in the message function, we must choose an additive function. A simple possible choice is vector addition: $\mu_{\text{ADD}}(\mathbf{h}_u, \mathbf{a}_r) = \mathbf{h}_u + \mathbf{a}_r$. However, additive update functions of this form have the problem that under a non-weighted sum aggregator, the relation vectors are interchangeable. It is not clear whether simply using an attention-based aggregator can resolve this problem completely since (subsets of) the attention weights can be close to each other. This leads us to the additional requirement of non-interchangeability of relations: $\forall \mathbf{a}, \mathbf{b}, \mathbf{c}, \mathbf{d} \in \mathbb{R} : \mathbf{a} \neq \mathbf{c} \wedge \mathbf{b} \neq \mathbf{d} \iff \mu(\mathbf{a}, \mathbf{b}) + \mu(\mathbf{c}, \mathbf{d}) \neq \mu(\mathbf{a}, \mathbf{d}) + \mu(\mathbf{c}, \mathbf{b})$. The third requirement is vector-based parameterization of relation types to combat over-parameterization in RGCN and GGNN.

We propose the following two-layer gated message function, which fulfills these requirements and can model complex interactions between node and edge features (unlike some simpler choices that

|  | L=5 | L=10 | L=15 | L=5 (N=16) |
|---|---|---|---|---|
| RGCN | $91.8 \pm 2.4$ | $53.5 \pm 4.9$ | $6.3 \pm 8.9$ | $47.5 \pm 1.8$ |
| GGNN | $69.7 \pm 3.3$ | $2.5 \pm 1.8$ | $3.3 \pm 2.4$ | $80.1 \pm 1.4$ |
| RGAT | $91.0 \pm 1.2$ | $75.1 \pm 3.2$ | $30.2 \pm 21.0$ | $41.0 \pm 3.3$ |
| GR-GAT (SGRU) | $\mathbf{98.6 \pm 1.0}$ | $\mathbf{96.7 \pm 0.7}$ | $\mathbf{93.2 \pm 2.7}$ | $\mathbf{95.9 \pm 4.2}$ |
| SGRU | $94.8 \pm 2.3$ | $89.9 \pm 2.0$ | $84.4 \pm 0.7$ | $92.9 \pm 4.0$ |
| GRU | $93.7 \pm 2.5$ | $9.3 \pm 3.7$ | $4.6 \pm 1.9$ | $81.1 \pm 3.5$ |

Table 1: Conditional recall results. The average accuracy on the test set is reported. L is the length of the generated sequences. The last column is a setting where the task length is 5 but a 16-layer network is used.

also fulfill the requirements for the message function).

$$\mathbf{z} = \mathsf{CELU}(\mathbf{b}_A + \mathbf{W}_A[\mathbf{h}_u; \mathbf{a}_r])) \ , \qquad \mathbf{m} = \sigma(\mathbf{b}_M + \mathbf{W}_M\mathbf{z}) \ , \qquad (7)$$

$$\mathbf{u} = \mathbf{b}_B + \mathbf{W}_B\mathbf{z} \ , \qquad \mu_{\mathrm{GCM}}(\mathbf{h}_u, r) = \mathbf{m} \odot \mathbf{h}_u + (1 - \mathbf{m}) \odot \mathbf{u} \ , \qquad (8)$$

where $\mathbf{a}_r$ is a vector $\in \mathbb{R}^D$ associated with the edge type from $u$ to $v$, and $\mathbf{W}_A$, $\mathbf{W}_B$, and $\mathbf{W}_M$ are trainable weight matrices (which have their corresponding biases $\mathbf{b}_A$, $\mathbf{b}_B$, and $\mathbf{b}_M$) that are shared between different edge types. See also Figure 1 for an illustration. CELU (Barron, 2017) is used as the activation function. Similarly to highway networks (Srivastava et al., 2015), $\mu_{\mathrm{GCM}}(\cdot)$ implements a gate that interpolates (based on $\mathbf{m}$) between the original node state $\mathbf{h}_u$ and the relation-aware update $\mathbf{u}$ using a two-layer feedforward network. See also Appendix A.3.

## 5 Experimental analysis

We wish to answer the following questions in our experiments:

Q.1 How well can the baselines learn simple rules for a sequence classification task? (5.1)

Q.2 How does the proposed model compare to the baselines on a synthetic node classification task requiring long-range pattern learning? (5.2)

Q.3 Are all the proposed changes necessary to achieve the best performance? (5.2)

Q.4 How does the proposed method perform in a shallow setup on entity classification using real-world knowledge graph data? (5.3)

Q.5 Does the performance degrade on entity classification when deeper networks are used? (5.3)

The following baselines are used: (1) RGCN (Schlichtkrull et al., 2018), (2) GGNN (Li et al., 2015) and (3) RGAT (Busbridge et al., 2019). We implement our RGCN and GGNN baselines based on the code provided by the DGL framework. For GGNN, the parameters between layers are shared. We found several relational variants of the GAT in the literature (Busbridge et al., 2019; Brockschmidt, 2019; Sinha et al., 2019) and chose to adapt (Busbridge et al., 2019) for the attention-based baseline (RGAT). See Appendix A.2 for an elaborate description of the used RGAT and other details regarding memory-efficiency.

The following ablations of our method (GR-GAT) are considered in the experiments: (1) "GR-GAT ($X$), mean agg." replaces the attention based aggregation function with a simple mean (equivalent to uniform attention), (2) "GR-GAT ($X$), edge-MM" replaces $\mu_{\mathrm{GCM}}$ with $\mu_{\mathrm{MM}}$, and (3) "GR-GAT ($X$), val. transf." uses a linear layer to transform the value while using attention, like in transformers. The $X$ in GR-GAT($X$) is the update function and can be one of the following: *Ident* (identity function), *GRU* (Gated Recurrent Unit), or *SGRU* (see Section 4.1). If $X$ is omitted, the *SGRU* has been used. Weight are always shared between all layers of GR-GAT.

### 5.1 Conditional Recall

In this experiment, we aim to answer Q.1 and show that the different popular relational GNNs are not able to solve a simple sequence classification task that involves remembering one of the symbols in

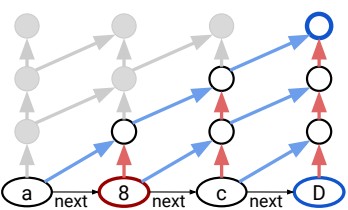

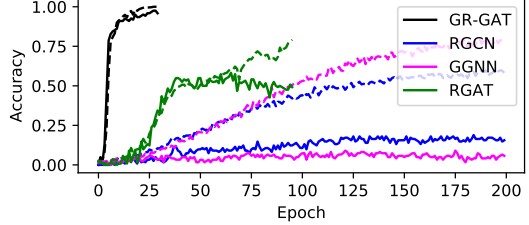

Figure 3: The input graph (ovals) for an example of the Conditional Recall task and the computational graph for 3 GNN layers.

Figure 4: Training (dashed lines) and validation (full lines) accuracies per epoch for the best runs for different GNN types on the Conditional Recall task (L=15).

a sequence according to a set of simple rules. Here we deliberately choose the following setup to measure how well the different GNNs can learn over a larger numbers of hops in a very simple graph.

We define a sequence classification task where given a sequence of characters $[x_1, \ldots, x_N]$, the model is asked to predict the correct class based on the representation of the last node (corresponding to input $x_N$). The input sequences consist of strings of letters and digits of a given length (which was varied between different experiments). The class of the sequence is determined by the following rules: (i) if there is a digit in the sequence, the first digit corresponds to the class label; (ii) otherwise, if there is an upper case character, the first upper case character is the class label; (iii) otherwise, the class is given by the first character in the sequence. Some examples are: "**a**bcdefg" → "a", "abc**D**efg" → "D", "abcd**3**Fg" → "3", "abCd**3**fg" → "3". Twenty examples were generated per output class for a total of 1220 examples and the data was split in 80/10/10 train/validation/test splits. For more details about the experimental setup for this task see Appendix A.5.

The input sequences are transformed into graphs by (i) creating a node for every character of the sequence and (ii) adding edges labeled using the `next` type between every adjacent element in the sequence. Given these edges, the GNN has to use at least $N$ layers/steps in order to propagate information from the first node to the last node. The number of layers is always set to $N + 1$. The readout for prediction is done by taking the representation of the last node in the sequence.

### 5.1.1 RESULTS AND DISCUSSION:

The results for the different GNNs and varying sequence lengths ($L$) are shown in Table 1 (Note that all examples have the same, specified length in one experiment). While all baselines reach acceptable accuracy for length 5, their accuracy severely suffers when the length is increased. In contrast, our method solves the task well for all lengths. Figure 4 reports the training behaviour in terms of training and validation set accuracies per epoch for the best runs of all considered GNNs, showing that the proposed GR-GAT model achieves much faster convergence and superior generalization on this task when compared to the baselines.

Additional experiments are conducted to directly compare SGRU and GRU. Here, the message function as well as the attention mechanism from GR-GAT were disabled, and self-edges were removed. The update function (SGRU or GRU) receives the neighbourhood aggregating vector as the input and the lower-level features of a node as the state. Clearly, the SGRU performs better, confirming our claims. Note that in this task, there is only one real edge type and every node has at most one neighbour so the message and aggregation functions are unnecessary.

Surprisingly, the GGNN severely underperforms compared to the skip-less RGCN and RGAT. We suspect that, because of the gated skip connections to lower-layer node representations, it is easier for the GGNN to memorize combinations of nearby nodes rather than learning meaningful patterns. Additional experiments on a modified task were conducted to verify the GGNN, for which the results are shown in the last column of Table 1. The modified task follows the same rules but uses short sequences (L=5) and a large number of layers (N=16). Even though many layers are used, the task thus requires looking over at most 4 hops in the graph. As we can see, the GGNN performs better than the other networks without depth-wise skip connections.

It is important to keep in mind that the computational graph for a GNN applied to this graph is different from an RNN applied to the same sequence. The "update distance" (the number of RNN or GNN updates performed) between the last state $h_T$ of an RNN and a token $x_{T-k}$ (from $k$ timesteps before) is $k$. In contrast, as illustrated in Figure 3, the update distance between the top-level features of any node and the initial features of any node is always equal to the number of layers in the GNN. In the figure, the final vector of the blue node is used for prediction and the red node specifies the desired output. Red arrows are "vertical" self-messages and blue arrows "horizontal" messages to neighbours. In the case of an RGCN, the neighbours and the node itself are treated equally, while the GGNN provides a skip connection to previous node features, enabling easier backpropagation "vertically" within the node. However, this does not improve backpropagation to distant nodes in the graph (see also Appendix A.4).

We also inspected the gradients w.r.t. the input vectors and experimented with different initializations for the edge type matrices of the baselines. We found that the initialization indeed strongly affects the gradient magnitudes and affects training behaviour. However, even when the weights are initialized correctly and the initial gradients are not small, the baselines are still not able to solve the task well for larger task lengths. We believe that our proposed changes encourage more effective weight sharing across nodes and layers and may benefit from other effects that make it easier for the network to generalize. This is similar to the findings of He et al. (2016).

## 5.2 TREE MAX

In this experiment, we aim to answer Q.2 and Q.3 using a synthetic node classification task on trees. The input graphs are trees with nodes labeled with random integers between 1 and 100. The expected output labels for the node classification task are defined as the largest value of all the descendants of a node and the node itself. The graphs contain edges from a parent to its children, and from children to their parent, as well as self-edges. We use numbered child edges and child-of edges, for example, :CHILD-1-OF for the edge going from the first child of a node to its parent.

As an example of this task, consider the following input tree: *(1 (2 (3 ) (4 )) (5 (6 ) (7 (8 ) (9 ) (10 ))))*. The nodes in the tree should be labeled with the value of their highest-valued descendant. Thus, the tree with its output labels will become: *(10 (4 (3 ) (4 )) (10 (6 ) (10 (8 ) (9 ) (10 ))))*. In this case, to correctly predict the output label of the root node (1), the GNN must handle three hops. Please see Figure 6 in Appendix A.6 for an illustration of the graph constructed for this example.

A total of 800 examples are generated, each containing one tree with a randomly chosen depth between 5 and 17. The largest generated trees contained more than 200 nodes. 17-layer GNNs (and fewer layers for the baselines) are used in the experiments. A 50/25/25% training/validation/test split is used. Instead of training and evaluating with the expected labels of all nodes, we choose a setting where the labels are omitted of node for which fewer hops must be resolved. Of the remaining nodes, only a random 50% selection retain their labels. Unlabeled nodes are not used in loss or metric computation. More details about the data and the experimental setup can be found in Appendix A.7.

### 5.2.1 RESULTS AND DISCUSSION:

As shown in Table 2, the best node-level and graph-level[2] accuracies were obtained using the proposed GR-GAT model. Using the SGRU doesn't bring benefit compared to having no update function (Identity) in this task. A possible explanation is that the task is retrieval-oriented and can thus easily be solved using just the attention mechanism.

Concerning the claimed improvements of SGRU vs a normal GRU, we can see that GRU is the worst among the three tested options (SGRU, GRU or Identity) by a large margin. The ablation study further indicates that the gated edge function (Section 4.3) and adapted attention mechanism (Section 4.2) are essential for achieving the best performance since the "edge-MM" and "val. transf." variants reach significantly worse performance.

Even though using no update function works well in this task, we believe the SGRU can be useful since it provides an alternative and more fine-grained mechanism for focusing on the previous node

---

[2]The graph-level accuracy is 100% for an example only if all labeled nodes in the graph have been classified correctly, and is 0% otherwise

|  | Node-level | Graph-level |
|---|---|---|
| RGCN | $63.4 \pm 1.1$ | $35.2 \pm 3.1$ |
| GGNN | $21.0 \pm 2.4$ | $7.7 \pm 2.5$ |
| RGAT | $45.6 \pm 5.5$ | $17.3 \pm 4.6$ |
| GR-GAT (SGRU) | $93.0 \pm 0.9$ | $82.5 \pm 1.1$ |
| GR-GAT (Ident) | $93.1 \pm 0.5$ | $83.0 \pm 0.7$ |
| GR-GAT (GRU) | $33.3 \pm 23.5$ | $20.4 \pm 23.2$ |
| GR-GAT (SGRU), mean agg. | $72.0 \pm 4.8$ | $48.5 \pm 5.9$ |
| GR-GAT (SGRU), edge-MM | $90.3 \pm 0.3$ | $77.0 \pm 0.8$ |
| GR-GAT (Ident), edge-MM | $50.5 \pm 18.5$ | $29.2 \pm 14.7$ |
| GR-GAT (Ident), val. transf. | $83.1 \pm 3.2$ | $65.3 \pm 4.3$ |

Table 2: Node and graph-level accuracies over the test set of the Tree Max task. Top part: baselines. Middle part: different update functions. Bottom part: other ablations. Note that the high variance for the *GR-GAT (GRU)* ablation is caused by one of the seven seeds resulting in much higher accuracy than the others.

| RGCN | 89.3 |
|---|---|
| SynGCN | $86.2 \pm 1.9$ |
| WGCN | $90.2 \pm 0.9$ |
| CompGCN | $\mathbf{90.6 \pm 0.2}$ |
| GR-GAT | $89.7 \pm 0.6$ |

|  | L=2 (2 hops) | L=10 (2 hops) | L=10 (subsample) |
|---|---|---|---|
| RGCN | 95.8 (*) | $88.0 \pm 1.3$ | $89.8 \pm 6.9$ |
| GGNN | $94.4 \pm 2.2$ | $92.6 \pm 1.3$ | $95.4 \pm 2.6$ |
| GR-GAT | $96.1 \pm 1.4$ | $92.6 \pm 1.3$ | $93.2 \pm 3.9$ |
| (Ident) | $\mathbf{96.3 \pm 1.3}$ | $92.6 \pm 2.6$ | $\mathbf{96.3 \pm 1.3}$ |

Table 3: Accuracy on the test set of the AM dataset.

Table 4: Accuracy on test set of AIFB for different data settings (see text). (*) - number taken from Schlichtkrull et al. (2018).

features (whereas the attention mechanism uses a self-edge relation when attending to the previous node features and the granularity is fixed to the number of heads). This can also be seen in the ablation study, where the SGRU variant performs much better with the matrix multiplication based edge function $\mu_{\mathrm{MM}}$ ("edge-MM" in the table) than the variant without an update function.

## 5.3 Entity classification using real-world data

To answer Q.4 and Q.5, we run experiments on real-world data for the node classification tasks, where a certain property about an entity (which are nodes in the graph) must be predicted. It is not clear whether this dataset could benefit from a GNN that is able to model longer patterns so we do not expect to see improvements using the original data. With these entity classification experiments, we mainly want to verify that our proposed model does not severely underperform compared to previously proposed relation-aware GNNs in a shallow setup and study the performance of our model when more layers are used.

The experimental setup is the same as in Schlichtkrull et al. (2018), who use previously proposed datasets for entity classification in knowledge graphs. The same preprocessed datasets and train/test splits are used here. However, compared to Schlichtkrull et al. (2018), who fit the entire dataset in memory, we implement subsampled mini-batch training. For more details, see Appendix A.8. Experiments are performed using two layers (similarly to Schlichtkrull et al. (2018)), for which the results are reported in Table 3 for the AM dataset and in the first column of Table 4 for the AIFB dataset. Further experiments are performed using the AIFB dataset to measure how much GR-GAT degrades when more layers are used, which is shown in the rest of Table 4.

### 5.3.1 Results and Discussion

The results in Tables 3 and the first column of Table 4 show that our method is competitive with the previously proposed relation-aware GNNs in a two-layer setup ($L = 2$). The second column of Table 4 shows that GR-GAT degrades less strongly than RGCN when ten layers ($L = 10$) are used on the same two-hop subgraph ("2 hops") as in the first column. The GGNN, which has vertical

skip-connections and also shares parameters between layers, is similarly more resilient to the increase in layer count. When a sampled subgraph is used that is built by random walks from the labeled node (which goes beyond two hops; see also Appendix A.8), which is the setting reported in the last column of Table 4, we see some small improvement from all methods compared to a 10-layer network applied on the two-hop subgraphs. However, there is still no improvement compared to the 2-layer GNN applied to a 2-hop subgraph. It is unclear whether the dataset could benefit from long-range pattern learning.

## 6 RELATED WORK

Li et al. (2019a) investigate the GCN equivalents of ResNet (He et al., 2016) and DenseNet (Huang et al., 2017), evaluated on point cloud semantic segmentation, as well as dilated aggregation. Residual-GCNs and Dense-GCNs (Li et al., 2019a), Highway GCN (Rahimi et al., 2018), Column Networks (Pham et al., 2017) and Li et al. (2020) are similar to the GGNN (Li et al., 2015) in using gated, residual or concatenated skip connections to previous nodes states. Xu et al. (2018) propose to add skip connections from all layers straight to the output layer to combat oversmoothing. The neighbourhood dilation proposed by Li et al. (2019a) for point cloud semantic segmentation increases the receptive field of the network without requiring the information to pass through other nodes. However, it is not clear how to apply this technique to multi-relational graphs. Additionally, different normalization techniques (Zhou et al., 2020; Zhao & Akoglu, 2020; Li et al., 2020) have very recently been proposed that improve the training of deep GNNs. AGGCN (Guo et al., 2019) transforms a graph into a fully connected graph and uses self-attention and thus, similarly to transformers (Vaswani et al., 2017), benefits from direct access to distant nodes. However, this leads to quadratic complexity in the number of nodes (due to attention in a fully connected graph). In contrast to previous work, we focus on improving the communication between neighbours in GNNs for multi-relational graphs.

A family of methods closely related to GNNs are TreeLSTMs (Tai et al., 2015) and GraphLSTMs (Peng et al., 2017; Liang et al., 2016; Song et al., 2018; Bresson & Laurent, 2017). Some versions of these methods have similar beneficial properties regarding backpropagation to distant nodes but are not suited for use in a general graph setup. Song et al. (2018) adapt Graph LSTMs into a message passing network, however, their implementation of the LSTM-based update suffers from the same issues as the GGNN. Compared to our proposed method, GraphLSTMs use independent forget gates for every neighbour instead of attention. Song et al. (2018) uses a global forget gate, which reduces the ability of the model to focus on certain nodes. The mixing gates in our SGRU bear some similarity to those in the Sentence State LSTM (Zhang et al., 2018). However, the original formulation of the Sentence State LSTM is not applicable for graphs in general. Fi-GNN (Li et al., 2019b), similarly to a basic version of our work, uses a GGNN with attention-based aggregation. However, it lacks our modifications and investigation to improve long-range pattern learning.

## 7 CONCLUSION

In this work, we proposed a novel GNN architecture that can handle long-term dependencies in multi-relational graphs better than commonly used GNN architectures. Experiments on synthetic tasks show that popular GNNs are bad at learning long-range patterns, and are outperformed by the proposed method (GR-GAT). Real-world experiments show that our proposed method is at least competitive with previous work.An ablation study indicates that the proposed changes w.r.t. the GAT and the GGNN are essential. The SGRU, which beats the GRU, is not absolutely necessary to have as the update function because the attention mechanism can mostly compensate for it while avoiding introducing additional parameters. However, we believe that the SGRU could still be useful as an orthogonal mechanism to control node representations and should be explored in applications. In future work, we plan to investigate additional model improvements and training techniques to further improve training of long-range relational GNNs.

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

# A   APPENDIX

## A.1   ATTENTION-BASED AGGREGATOR

The query and key vectors for each attention head $h$ (of in total $H$ heads), $\mathbf{q}_v^{(h)} \in \mathbb{R}^D$ for node $v$ and $\mathbf{k}_e^{(h)} \in \mathbb{R}^D$ for an edge $e = (u, v, r)$ ending in $v$, respectively, are computed using head-specific linear transformations parameterized by $\mathbf{Q}^{(h)} \in \mathbb{R}^{D/H \times D}$ and $\mathbf{K}^{(h)} \in \mathbb{R}^{D/H \times 2D}$:

$$\mathbf{q}_v^{(h)} = \mathbf{Q}^{(h)} \mathbf{h}_v \ , \qquad\qquad \mathbf{k}_e^{(h)} = \mathbf{K}^{(h)}[\mu(\mathbf{h}_u, r); \mathbf{a}_r] \ , \qquad\qquad (9)$$

where $[\cdot; \cdot]$ denotes the (vertical) concatenation of two (column) vectors and $\mathbf{a}_r$ is the same edge type-specific parameter vector that is used in Eqs. 7-8. Note that Eq. 9 is slightly different from a standard attention mechanism, which would compute key vectors as $\mathbf{K}^{(h)} \mu(\mathbf{h}_u, r)$. We expect that including the edge type vector $\mathbf{a}_r$ directly in the keys will make it easier to learn to place attention.

The attention score $w_e^{(h)}$ for an edge $e$ and head $h$ is computed by a scaled dot product and then normalized to obtain $\alpha^{(h)}$ as usual: $w_e^{(h)} = \frac{\mathbf{q}_v^{(h)} \cdot \mathbf{k}_e^{(h)}}{\sqrt{D/H}}$ and $\alpha_e^{(h)} = \frac{\exp(w_e^{(h)})}{\sum_{e^* \in \mathcal{E}(\cdot, v)} \exp(w_{e^*}^{(h)})}$.

Unlike multi-head attention in transformers and previous work (Veličković et al., 2018; Busbridge et al., 2019), we do not use linear transformations to obtain the value vectors. Instead, the vector $\mu(\mathbf{h}_u, r) = \boldsymbol{\mu}_e = [\boldsymbol{\mu}_e^{(1)}, \ldots, \boldsymbol{\mu}_e^{(H)}]$ is split in $H$ equally sized chunks and the $h$-th part $\boldsymbol{\mu}_e^{(h)} \in \mathbb{R}^{D/H}$ is taken as value vector for the $h$-th head. The attention scores $\alpha_e^{(h)}$ and value vectors $\boldsymbol{\mu}_e^{(h)}$ are then used to compute the neighbourhood aggregation vector $\overline{\mathbf{h}}_v^{(h)}$ for head $h$ as a weighted sum. The full neighbourhood aggregation vector $\overline{\mathbf{h}}_v$ is then given as a concatenation over all heads $h$:

$$\overline{\mathbf{h}}_v^{(h)} = \sum_{e \in \mathcal{E}(\cdot, v)} \alpha_e^{(h)} \boldsymbol{\mu}_e^{(h)} \ , \qquad\qquad \overline{\mathbf{h}}_v = [\overline{\mathbf{h}}_v^{(1)}; \ldots; \overline{\mathbf{h}}_v^{(H)}] \ . \qquad\qquad (10)$$

## A.2   BASELINES

### A.2.1   A NOTE ON MEMORY-EFFICIENCY OF THE BASELINES

When experimenting with the baselines (RGCN, GGNN and RGAT), we quickly ran out of memory for larger hidden dimensions of the GNN and larger numbers of layers. This is caused by the relation matrices, which have $O(n^2)$ parameters for every relation. In the implementations we used, the relations were indexed from a tensor and have to be retained in memory for every edge. While more memory-efficient implementations are possible, the existing ones sacrifice execution time, which can increase significantly with a large number of different relations.

Gradient accumulation significantly slows down training. As a simple fix to maintain reasonable training speed for our experiments, we replace $\mu_{\mathrm{MM}}(\cdot)$ in our RGCN and GGNN with $\mu_{\mathrm{MM-red}}(\mathbf{h}_u) = W_B W_r^* W_A \mathbf{h}_u$, where $W_r^*$ is an edge type specific square matrix of lower dimensionality than $W_r$ and $W_A$ and $W_B$ are matrices projecting into and out of $W_r^*$'s dimensionality that are shared for all edge types. Note that we did this only for the Tree Max experiments.

### A.2.2   RGAT BASELINE

Like in (Busbridge et al., 2019), we use relation-specific transformation matrices $\mathbf{W}^{(r)}$ and relation and head-specific query and key matrices $\mathbf{Q}^{(r,h)}$ and $\mathbf{K}^{(r,h)}$.

First, we define relation-dependent representations for a node, which is computed based on its current state $\mathbf{z}^{(k-1)}$:

$$\mathbf{g}_i^{(r)} = \mathbf{W}^{(r)} \mathbf{z}_i^{(k-1)} \ . \qquad\qquad (11)$$

Subsequently, for every head $h$, we define the relation-specific query, key and value projections as:

$$\mathbf{q}_i^{(h,r)} = \mathbf{Q}^{(h,r)} \mathbf{g}_i^{(r)} \qquad\qquad (12)$$

$$\mathbf{k}_i^{(h,r)} = \mathbf{K}^{(h,r)} \mathbf{g}_i^{(r)} \qquad\qquad (13)$$

$$\mathbf{v}_i^{(h,r)} = \mathbf{V}^{(h,r)} \mathbf{z}_i^{(k-1)} \qquad\qquad (14)$$

We compute attention between messages $\boldsymbol{\mu}_x$, where $x$ indexes over all edges in the input graph. First, we compute the attention score for a message as in (Vaswani et al., 2017):

$$s_{\boldsymbol{\mu}_x}^{(h)} = \mathbf{q}_i^{(h,r)} \cdot \mathbf{k}_j^{(h,r)} \quad, \tag{15}$$

where $\boldsymbol{\mu}_x$ is the message sent along an edge $j \rightarrow i$ labeled by the relation $r$. Please note that a node $i$ may receive multiple messages from a node $j$, and that those messages could contain the same relation.

The scores are normalized over all messages that node $i$ receives:

$$\alpha_{\boldsymbol{\mu}_x}^{(h)} = \frac{s_{\boldsymbol{\mu}_x}^{(h)}}{\sum_{\boldsymbol{\mu}_{x'}} s_{\boldsymbol{\mu}_{x'}}^{(h)}} \tag{16}$$

The normalized scores are used to compute a summary:

$$\mathbf{z}_i^{(h,k)} = \sum_{\boldsymbol{\mu}_x} \mathbf{v}_j^{(h,r)} \alpha_{\boldsymbol{\mu}_x}^{(h)} \quad, \tag{17}$$

where $r$ is the relation of $\boldsymbol{\mu}_x$ and $j$ is the source node id and $i$ is the target node id of message $\boldsymbol{\mu}_x$.

The updated representation for node $i$ is then the concatenation over all heads, fed through the activation function $\sigma$:

$$\mathbf{z}_i^{(k)} = \sigma([\mathbf{z}_i^{(0,k)}, \dots, \mathbf{z}_i^{(H,k)}]) \quad. \tag{18}$$

We experiment with a ReLU, as well as with a linear $\sigma$.

### A.3 NOTES ON RELATION-AWARE MESSAGE FUNCTIONS.

Many different message function implementations are possible that take into account the relation connecting a node to a neighbour. The default message function used in earlier literature is simply a matrix multiplication:

$$\mu_{\mathrm{MM}}(\mathbf{h}_u, r) = \mathbf{W}_r \mathbf{h}_u \tag{19}$$

This can easily lead to overparameterization and leads to difficulties implementing the GNNs in a both memory-efficient and computationally efficient way.

In CompGCN, Vashishth et al. (2020) propose the following function:

$$\mu_{\mathrm{CompGCN}}(\mathbf{h}_u, r) = \mathbf{W}_{\lambda(r)} \phi(\mathbf{h}_u, \mathbf{a}_r) \quad, \tag{20}$$

where $\mathbf{W}_{\lambda(r)}$ is one of the three matrices: one for forward relations, one for reverse and one for self-edges, and $\phi(\mathbf{h}_u, \mathbf{a}_r)$ is a composition function that combines the neighbour state and relation vector. For the composition function, the authors explore functions inspired by knowledge graph embedding literature: subtraction ($\mathbf{h}_u - \mathbf{a}_r$), multiplication or circular-correlation.

While this provides an alternative solution to over-parameterizing relations, and enables more efficient implementation, it does not improve gradient behavior since the backpropagation paths to the lower node features as well as to distant nodes still contain linear transformation and non-linearities.

To improve backpropagation, we can make use of (gated or residual) skip connections. The residual version would have the following general form:

$$\mu_{\mathrm{Res}}(\mathbf{h}_u, r) = \mu_{\mathrm{X}}(\mathbf{h}_u, r) + \mathbf{h}_u \quad, \tag{21}$$

where $\mu_{\mathrm{X}}$ is some function. Similarly to transformers and ResNets, we choose to use a two-layer MLP, which is more expressive than $\mu_{\mathrm{MM}}$, and unlike the composition-based message functions in Vashishth et al. (2020), can effortlessly incorporate additional edge features that are not based on the relation embeddings.

We chose the gated skip connection described in the text over the simple residual one because it enables better control of the additive contribution of the current layer and the residual version did not perform better in our early experiments. We leave a further investigation of residuals to replace the gated skip connections for future work.

### A.3.1 A NOTE ON INTERCHANGEABILITY OF RELATIONS

Note that the simple choice of adding a relation vector, similarly to CompGCN's subtraction version, might pose problems since it loses information about which relation was associated with which neighbour:

$$(\mathbf{h}_u + \mathbf{a}_r) + (\mathbf{h}_w + \mathbf{a}_r) = (\mathbf{h}_w + \mathbf{a}_r) + (\mathbf{h}_u + \mathbf{a}_r) \ . \tag{22}$$

This is undesireable since it limits the expressive power of the network when using a sum aggregator. It is not clear to us how much this would affect models using attention-based aggregation, however, we prefer to avoid this since the attention-based aggregator may behave like a uniform sum, especially throughout the early stages of training.

This leads to the following condition that we impose on the message function: The message function $\mu$ must be chosen such that

$$\mu(\mathbf{a}, b) + \mu(\mathbf{c}, d) \neq \mu(\mathbf{a}, d) + \mu(\mathbf{c}, b) \tag{23}$$

iff $a \neq c$ and $b \neq d$.

The gated message function $\mu_{\mathrm{GCM}}$ proposed earlier in our work satisfies this condition.

### A.4 BACKPROPAGATION TO DISTANT NODES IN GATED GRAPH NEURAL NETWORKS

The GGNN (Li et al., 2015) implements the update function $\phi(\cdot)$ based on Gated Recurrent Units (GRUs) (Cho et al.):

$$\mathbf{r}^{(k)} = \sigma(W_r \overline{\mathbf{h}}_v^{(k-1)} + U_r \mathbf{h}_v^{(k-1)} + b_r) \tag{24}$$

$$\mathbf{z}^{(k)} = \sigma(W_z \overline{\mathbf{h}}_v^{(k-1)} + U_z \mathbf{h}_v^{(k-1)} + b_z) \tag{25}$$

$$\hat{\mathbf{h}}_v^{(k)} = \tanh(W \overline{\mathbf{h}}_v^{(k-1)} + U(\mathbf{h}_v^{(k-1)} \odot \mathbf{r}^{(k)})) \tag{26}$$

$$\mathbf{h}_v^{(k)} = (1 - \mathbf{z}^{(k)}) \odot \mathbf{h}_v^{(k-1)} + \mathbf{z}^{(k)} \odot \hat{\mathbf{h}}_v^{(k)} \ , \tag{27}$$

where $\overline{\mathbf{h}}_v^{(k-1)} = \gamma(\{\mu(\mathbf{h}_u^{(k-1)}, r)\}_{(u,v,r)\in\mathcal{E}(\cdot,v)})$ is the vector representing the aggregated neighbourhood of node $v$.

Consider a node classification task, where the top-level state $\mathbf{h}_v^{(K)}$ of node $v$ is fed into a classifier that produces a loss $\mathcal{L}_v$. To compute the gradient $\nabla_{\mathbf{w}} \mathcal{L}_v = \dfrac{\partial \mathcal{L}_v}{\partial \mathbf{w}}$, where $\mathbf{w}$ are the parameters of the model, we apply the chain rule as follows (here we ignore other nodes and only take into account only node $v$'s contribution to the gradient):

$$\nabla_{\mathbf{w}} \mathcal{L}_v = \frac{\partial \mathcal{L}_v}{\partial \mathbf{w}} = \sum_{k=0}^{K} \frac{\partial \mathcal{L}_v}{\partial \mathbf{h}_v^{(k)}} \frac{\partial \mathbf{h}_v^{(k)}}{\partial \mathbf{w}} \tag{28}$$

with

$$\frac{\partial \mathcal{L}_v}{\partial \mathbf{h}_v^{(k)}} = \frac{\partial \mathcal{L}_v}{\partial \mathbf{h}_v^{(K)}} \prod_{j=k+1}^{K} \frac{\partial \mathbf{h}_v^{(k)}}{\partial \mathbf{h}_v^{(k-1)}} \tag{29}$$

Given the gated update equations of the GRU, the partial derivatives $\dfrac{\partial \mathbf{h}_v^{(k)}}{\partial \mathbf{h}_v^{(k-1)}}$ of the updated node representation w.r.t. the previous node representation are:

$$\frac{\partial \mathbf{h}_v^{(k)}}{\partial \mathbf{h}_v^{(k-1)}} = (1 - \mathbf{z}) \odot \mathbf{I} + \mathbf{z} \odot \frac{\partial \hat{\mathbf{h}}_v^{(k)}}{\partial \mathbf{h}_v^{(k-1)}} + \frac{\partial(-\mathbf{z})}{\partial \mathbf{h}_v^{(k-1)}} \odot \mathbf{h}_v^{(k-1)} + \frac{\partial \mathbf{z}}{\partial \mathbf{h}_v^{(k-1)}} \odot \hat{\mathbf{h}}_v^{(k)} \tag{30}$$

The closer $\mathbf{z}$ is to a zero vector, the closer the partial derivatives across layers $\dfrac{\partial \mathbf{h}_v^{(k)}}{\partial \mathbf{h}_v^{(k-1)}}$ get to the identity matrix $\mathbf{I}$ and the more the following holds:

$$\frac{\partial \mathcal{L}_v}{\partial \mathbf{h}_v^{(k)}} \approx \frac{\partial \mathcal{L}_v}{\partial \mathbf{h}_v^{(K)}} \quad , \tag{31}$$

which has the effect that all the GGNN layers are skipped. In practice, the gate $\mathbf{z}$ will take on different values throughout training, but the additive update still allows to retain some contribution of the original gradient $\dfrac{\partial \mathcal{L}_v}{\partial \mathbf{h}_v^{(K)}}$ deeper into the network.

On the other hand, consider a (K-1)-layer GGNN applied to a graph representing a sequence of length K. The sequence is represented as a graph according to the same rules as described in Section 5.1: the nodes are connected using only `next`-edges. Thus, only left-to-right propagation in the graph can be performed, and every node has only one incoming message, except the first, which has none. This is the setup that maximizes the ratio of graph diameter per number of nodes and edges.

When a (K-1)-layer GGNN is applied to such a graph of K nodes, there is only one backpropagation path from node K to node 0: $\dfrac{\partial \mathbf{h}_K^{(K)}}{\partial \mathbf{h}_0^{(0)}}$, which is also the longest backpropagation path possible in this graph:

$$\frac{\partial \mathbf{h}_K^{(K)}}{\partial \mathbf{h}_0^{(0)}} = \prod_{k=1}^{K} \frac{\partial \mathbf{h}_k^{(k)}}{\partial \mathbf{h}_{k-1}^{(k-1)}} \tag{32}$$

Ignoring the used message function $\mu_{\mathrm{MM}}$ and considering every node has only one neighbour, the neighbourhood aggregation vector is simply the neighbour node vector: $\overline{\mathbf{h}}_v^{(k-1)} = \mathbf{h}_{k-1}^{(k-1)}$.

Plugging this into the GGNN equations (Eqs. 24-27) gives:

$$\frac{\partial \mathbf{h}_k^{(k)}}{\partial \mathbf{h}_{k-1}^{(k-1)}} = \frac{\partial \mathbf{z}^{(k)} \tanh(W \mathbf{h}_{k-1}^{(k-1)})}{\partial \mathbf{h}_{k-1}^{(k-1)}} \tag{33}$$

$$= \tanh(W \mathbf{h}_{k-1}^{(k-1)}) \frac{\partial \mathbf{z}^{(k)}}{\partial \mathbf{h}_{k-1}^{(k-1)}} + \mathbf{z}^{(k)} \frac{\partial \tanh(W \mathbf{h}_{k-1}^{(k-1)})}{\partial \mathbf{h}_{k-1}^{(k-1)}} \tag{34}$$

$$= \tanh(W \mathbf{h}_{k-1}^{(k-1)}) \frac{\partial \sigma(W_z \mathbf{h}_{k-1}^{(k-1)})}{\partial \mathbf{h}_{k-1}^{(k-1)}} + \mathbf{z}^{(k)} \frac{\partial \tanh(W \mathbf{h}_{k-1}^{(k-1)})}{\partial \mathbf{h}_{k-1}^{(k-1)}} \tag{35}$$

Both terms contain a partial derivative of the form $\dfrac{\partial \sigma(W \mathbf{h}_{k-1}^{(k-1)})}{\partial \mathbf{h}_{k-1}^{(k-1)}}$, which is essentially identical to the terms found in the backpropagation equations for a vanilla RNN.

As in vanilla RNNs (without any gating like in GRUs and LSTMs), these terms can cause vanishing gradients due to repeated multiplication with the derivative of a $\sigma$ or $\tanh$ nonlinearity and vanishing or exploding gradients depending on the values of $W$ (which also change throughout training and thus could potentially reach unstable values).

In addition to the gradient-based argument outlined above, residual networks (as well as GRUs and LSTMs), benefit from better generalization abilities due to effects which were not formalized or deeply studied in the literature known to us. Intuitively, the argument for residual networks (He et al., 2016) is that the individual residual blocks can easily learn to *not* contribute if necessary, which enables to learn very deep architectures, and otherwise, they only need to model the function that models the *residual* error computed from the output of the previous layer. In addition, we believe that adding such skip connections enables more effective weight sharing across layers and nodes.

### A.5 CONDITIONAL RECALL EXPERIMENTAL DETAILS:

In our experiments for this task, we randomly explore the following hyperparameter values for all compared methods: (1) dimensionality of node feature vectors in $[150, 300]$ (larger values can cause

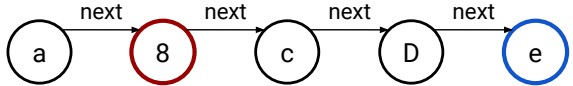

Figure 5: Example of input graph for the Conditional Recall task. The top-level state of the blue node is used for prediction. The red node specifies the desired output.

memory issues with the baselines), (2) dropout in $[0., 0.25, 0.5]$, (3) we set the dropout of embedding vectors to 0.1 (4) learning rate in $[0.001, 0.0005, 0.0001]$. For the baselines, we run for at least 200 epochs. For our GR-GAT, we reach high accuracy in 20 epochs and thus run only for a maximum of 50 epochs. Throughout all of the experiments, we use the Adam (Kingma & Ba, 2015) optimizer. We also use label smoothing with a factor 0.1 everywhere. For our experiments with task length 15, we use the best hyperparameters found for length 10 for all methods and also experiment with smaller dropout values.

For the final reported accuracies, we trained with the best hyperparameters and loaded the model with the best validation error and evaluated on the test set. This was repeated with three different seeds that were shared between the tested methods.

The initial node states are initialized by embedding the node type using a low-dimensional embedding matrix (dimensionality 20; for a vocabulary of 62 characters) and projecting the low-dimensional embeddings to the node state dimension using a learned transformation.

An example of a graph used in this task is given in Figure 5. The output in this example should be "8".

## A.6   TREE MAX EXAMPLE

See Figure 6 on page 17.

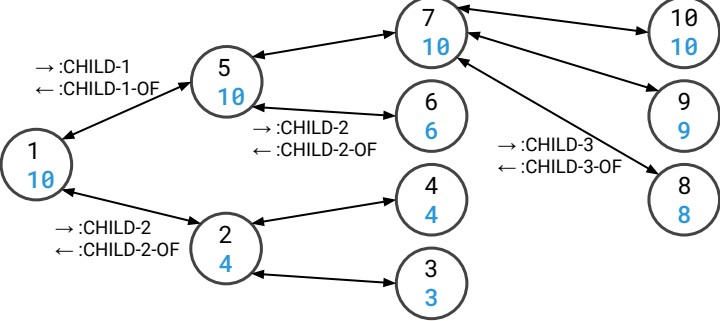

Figure 6: Example of input graph for the Tree Max task. The double-ended arrow between a node $v$ and a node $u$ represents two edges: one going from $v$ to $u$ and the other from $u$ to $v$. Not every arrow is labeled for a clearer presentation. The labels on the arrows indicate edge labels: the forward arrow $\rightarrow$ corresponds to the edge label for going from the parent to the child and $\leftarrow$ for the reverse. The input labels are the black numbers and the output labels are the blue numbers. Note that we use a semi-supervised version of this task, where we erase the output labels of a large portion of the nodes, i.e. ignore their output labels in training and testing.

## A.7   TREE MAX EXPERIMENTAL DETAILS:

During data generation, we first randomly pick a tree depth between five and fifteen. Then, we generate a tree, choosing between 0, 2 or 3 children for each node until we reach the chosen tree depth. The largest trees in our generated data contained over 200 nodes. The generated dataset contained 800 examples; we used a 50/25/25 train/validation/test split.

Not every node in the dataset is annotated. For every example, which consists of one graph, the nodes which are at most $N/2$ away from their "answer" node (which contains their output label) are ignored (not used in training or testing). Of the remaining nodes, which require to propagate information for at least $N/2$ to be solved, only 50% is retained for training and testing. Here, $N$ is the maximum depth of the tree. This setup defines a semi-supervised node classification setup. The reason for using the semi-supervised version instead of the previous version, which was fully supervised is that the task can be solved iteratively using immediate neighbour states only. Because of this, all methods reached very high node classification accuracy. In the semi-supervised version, the models have to learn what information to retain in unlabeled nodes, which are in between labeled nodes and their corresponding "answer" nodes.

For every model tested, we perform a hyperparameter search using all predefined seeds. For all models, we experiment only with dimensionality of 150 to avoid memory issues and ensure a fair comparison regarding representation dimensionality. Then we run the best hyperparameter setting with three different random seeds and report the test results in Table 2. Note that the same seeds are re-used for experiments for all models, and that every seed results in a different dataset being generated. We used early stopping and reloaded the best model based on element-wise accuracy on the validation set. Patience was set to 10 epochs but every experiment was run for a minimum of 50 epochs.

The models were evaluated using node-wise and graph-wise accuracy. The graph-level accuracy is 100% for an example only if all gold-labeled nodes in the graph have been classified correctly, and is 0% otherwise.

In our experiments, we randomly explore different combinations of different hyperparameter settings: dropout is selected from $\{0., 0.1, 0.25, 0.5\}$, dimensionality of node vectors is fixed to $150$ for all methods. Learning rate is chosen from $\{0.001, 0.000333, 0.0001\}$. For all baselines, the number of layers was chosen from $10, 17$. Note however that 10 layers is insufficient to perfectly solve the task.

### A.8 Entity Classification Experimental Details

We use the datasets as provided at `https://github.com/tkipf/relational-gcn`, removing some code for more memory efficient storage. The datasets contain graphs, where only a fraction of nodes carries a label that we should predict (we call these nodes labeled nodes). We train on the train portion as provided on the repository and test on the predefined test portion.

During experiments, we train with two different settings: (1) a fixed two-hop (subsampled) subgraph and (2) a fixed subsampled random walk subgraph.

In the first setting (two-hop subgraph), we retrieve for every labeled node that will be classified a two-hop, at most 6000 nodes (and 10000 edges) subgraph. During batching, these subgraphs are merged. Note that if the number of nodes (or edges) exceeds the maximum number of nodes (6000), we stop expanding. We experiments with the following hyperparameter ranges: (1) dropout from 0.25, 0.5, (2) embedding dropout from 0.25, 0.5, (3) dimension from 64, 96, (4) learning rate from 0.005, 0.001, 0.00025. We use at least 50 epochs and early stopping with patience set to 10 epochs to terminate validation runs earlier and save computation time. The number of heads in our GR-GAT method is set to four for all tried dimensions. During hyperparameter search, we run a random selection of all possible combinations with three different seeds, using a subset of the training set (80%) for training and another set (20%) for validation. After determining the best hyperparameters, we use them to train a final model on the entire training set and evaluate on the test set. The reported numbers use three different seeds.

In the second setting (random walk subgraph), we use random walks to generate the subgraph. For every labeled node, we precompute a subgraph of at most 2000 nodes and 3000 edges, using 5% reset probability in the random walk. During batching, these subgraphs are merged. Note that the subgraphs are precomputed once before training and fixed throughout, so the subgraphs do not get resampled at every epoch. We use the same set of seeds for the different methods we test to ensure that the methods have been tested on the same sets of data. We experiments with the following hyperparameter ranges: (1) dropout from 0., 0.1, 0.25, 0.5, (2) embedding dropout from 0., 0.1, 0.25, (3) dimension from 32, 64, (4) learning rate from 0.01, 0.005, 0.001, 0.00025. We use at least 50 epochs and early stopping with patience set to 10 epochs to terminate validation runs earlier and save

computation time. The number of heads in our GR-GAT method is set to 4 for all tried dimensions. Since GR-GAT (Ident) reached near 100% training accuracy with the chosen hyperparameters after 15 epochs, we ran it for 20 epochs during final tests. During hyperparameter search, we randomly run all possible combinations with three different seeds, using a subset of the training set (80%) for training and another set (20%) for validation. After determining the best hyperparameters, we use them to train a final model on the entire training set and evaluate on the test set.

In both settings, we erase the initial features (embeddings) of the nodes for which we make predictions by setting them to the same learnable vector that is reused everywhere.

