# OpenReview forum: "Gated Relational Graph Attention Networks"
_ICLR.cc/2021/Conference — Reject_

### Official Review · AnonReviewer2 · 2020-10-16
**Limited architecture novelty, no convincing performance on real tasks.**

**Rating:** 2
**Confidence:** 4

**Review:**

## Summary
This paper presents a graph attention architecture that captures long-range interactions. The novelties in the architectures are (1) vector-based parameterization of edge type in modeling message, (2) slight modification of graph attention (Section 3.2), and (3) GRU-based node update function. The experiments are primarily on synthetic tasks. However, it is unclear if modeling such long-range interaction is useful in real tasks. The paper fails to demonstrate convincing results on the real tasks of entity classification in knowledge graphs.

## Pros
1. Detailed architecture explanation.
2. Good performance on synthetic tasks.
3. Careful design of synthetic tasks.

## Cons:
1. The novelty of architecture is limited as detailed below.
- Vector-based parameterization of edge type in the relational graph has been commonly adopted in GNNs for molecular graphs (e.g., Eq (1) in https://arxiv.org/pdf/1709.04555.pdf), and not novel.
- The modifications of graph attention architecture are rather minor (removing a single linear transformation, adding edge type embedding in the key of the attention mechanism).
- GRU-based node update function is not empirically shown to be beneficial although being highly complicated.
2. Experiments are largely synthetic, and no convincing results are provided for the real datasets on entity classification in knowledge graphs. It is unclear if modeling long-range interaction is useful in practice. One domain long-range interaction could be useful is molecule classification, where you can treat molecular graphs as multi-relational graphs and those graphs tend to have large graph diameters. Many datasets are readily available [here](https://ogb.stanford.edu/docs/graphprop/).
3. Details of the real knowledge graph datasets (AIFB and AM) are not provided in the main texts.

---

> ### Author Response · Authors · 2020-11-18
> **Response to Reviewer 2**
>
> Thank you for taking the time to review our work and providing helpful comments.
> We are updating our paper using your comments and provide it to you as soon as possible.
>
> See below for detailed responses to the concerns you raised:
>
> (1) We agree that the changes we propose build on well-known principles in neural network design. However, the concrete method we propose is novel. Especially, to the best of our knowledge, it is the first that focuses on the problems associated with breadth-wise backpropagation. We are not aware of any existing architecture within the message-passing framework that would not have these problems.
> It seems like we did not do a good job of differentiating our design decisions from other works and motivating our design decisions but we will improve our explanations using the additional space in the elaborated version.
>
> (1a) Thank you for the paper, we will add it to our references. Based on your comments, we realized that the proposed message function has not been motivated properly. But we would like to highlight that our main contribution there is not vector-based parameterization, which has been used in several previous works, as we mention in the paper, but the improved backpropagation compared to the previously proposed models. However, with the extra space, we will more clearly explain the motivation.
>
> (1b) The changes are indeed minor. However, removing the value transformation seems to be essential. Without it, the entire method appears to significantly underperform.
>
> (1c) We will run additional experiments for the first task to study the SGRU in isolation.  We already ran some additional ablation comparisons on the second task where the SGRU provides improvement when coupled with \mu_MM. We will add these additional results to the updated version of our paper.
>
> (2) Finding a good dataset with long-range dependencies that has been used by previous work has been a major issue for this work and we thank you for suggesting the molecular datasets from OGB. We will be working on these experiments. We would also appreciate more dataset suggestions.
> One problem in general with graph classification tasks though, could be that the final graph pooling layer is already enough to capture distant interactions and thus, the changes we propose would not lead to improvement. This is also why we focused on node classification tasks. What do you think about this argument? Would you suggest any node classification datasets that could be useful here?
>
> (3) We will provide details about the datasets in text.

---

### Official Review · AnonReviewer3 · 2020-10-27
**Recommendation to reject**

**Rating:** 5
**Confidence:** 4

**Review:**

#####Summary#####

This paper proposes a new GNN model (GR-GAT) for multi-relational graphs. The proposed method has better ability of capturing the long-range information. Essentially, the proposed GR-GAT is modified from GAT so that it can apply to the multi-relational graphs. Since the modifications are common and frequently used techniques, the novelty of this work is not enough. Also, why these modifications can help to capture long-range information is not well explained in this paper. Overall, this work is ok but not good enough for ICLR.

#####Pros#####

(1) The experiments on synthetic are well designed and can show the power of the proposed GR-GAT.

(2) This paper studies a meaningful and challenging problem; that is capturing long-range information using GNNs.

#####Cons#####

(1) The novelty of the proposed method is not enough. Based on the description in Section 3, the proposed model is basically under the message passing framework (Eq (1)) and the concrete implementations of the three functions included in the message passing framework is not novel. For example, parameterizing message function using vector, aggregating neighborhood using attention, and updating using GRU are all popular used techniques in GNNs. Overall, this method is likely to be a combination of GAT and some basic techniques for multi-relational graphs.

(2) More importantly, the motivation is not explained clearly. The current version did not well explain why the proposed model can help to model long-range dependencies in Section 3.

(3)  To show the ability of capturing long-range dependencies, the experiments are only conducted on small synthetic datasets and specific defined tasks. It is not enough to show the ability of capturing long-range information. More experiments and comparisons on large real-world datasets should be considered.

#####Suggestions for improvement#####

(1)	I think the main issue of the current version is that our readers cannot tell the contributions of this paper from Section 3. We cannot find what is the key proposal of this paper and how this proposal can help to capture long-range information. I think if this point can be improved, the novelty and motivation of this paper can be clearer.


######

Update: After looking at the revised version, I would like to raise my score to 5 since the motivation is clearer.

---

> ### Author Response · Authors · 2020-11-18
> **Response to Reviewer 3**
>
> Thank you for taking the time to review our work and providing helpful comments.
> We are updating our paper using your comments and hope to provide it to you as soon as possible.
>
> See below for detailed responses to the concerns you raised:
>
> (1) We agree that the changes we propose build on well-known principles in neural network design. However, the concrete method we propose is novel. Especially, to the best of our knowledge, it is the first that focuses on the problems associated with breadth-wise backpropagation. We are not aware of any existing architecture within the message-passing framework that would not have these problems.
> It seems like we did not do a good job of differentiating our design decisions from other works and motivating our design decisions but we will improve our explanations using the additional space in the elaborated version.
> Please note that the proposed SGRU is different from the GRU typically used in GGNNs and actually aims to solve the shortcomings of GGNN’s GRU.
> Also, please note that in addition to vector-based parameterization, we use a message function that improves backpropagation compared to, for example, CompGCN. This property is more interesting to us than the fact that the relations are parameterized using vectors. We will improve the explanation in the paper to better convey this.
>
> (2) The explanation and motivation should indeed be improved, as other reviewers also pointed out, and will be a major focus point for the next version of the paper. Specifically, we will more clearly explain the problem we are addressing (improving breadth-wise backpropagation), reorganize the approach section and motivate the decisions more extensively. Currently, the motivation for the breadth-wise backpropagation is extensively discussed only for the GGNN, in appendix.
>
> (3) We agree that we need more experiments on real data with hopefully more conclusive results. We will try to run experiments on some molecular datasets from OGB. We would also appreciate any suggestions for additional synthetic and/or real tasks.

---

> > ### Comment · AnonReviewer3 · 2020-11-18
> > **Response to authors**
> >
> > Thank you for your response.
> >
> > Hope the future version of this paper could be improved, especially in the aspect of motivation and method description.

---

### Official Review · AnonReviewer1 · 2020-10-28
**Promising long-range tweaks to the relational GAT, but lacks clear motivation**

**Rating:** 4
**Confidence:** 4

**Review:**

The authors propose Gated Relational Graph Attention Nets (GR-GAT), a set of modifications to the GAT architecture in order to make them stronger under long-range relational reasoning, evaluating on various hand-crafted benchmarks as well as a real-world dataset previously explored in the area.

The GR-GAT tackles all aspects of the graph neural network pipeline:
- Message function that takes into account the edge type embedding, and features explicit gating over the sender node's past features, and usage of the CELU activation;
- Multi-head attentional aggregator which splits the value vectors into chunks, rather than replicating them;
- Symmetric approach to the GRU update rule, which factors both inputs into the gating.

The three approaches are potentially meaningful for ameliorating various issues with long-range reasoning, such as overfitting or vanishing gradients, and I think the methodology from this paper could be useful to GNN practitioners. However, the paper's current presentation and motivation does not feel suitable for a venue like ICLR, in my opinion.

My main concern is that the authors do not properly motivate most of their design choices, or properly ground them in a particular issue with (R)GATs or GGNNs. Many choices (to name a few: the gating message, the CELU activation, or the splitting value vector) are name-dropped in the paper, without properly explaining their significance in the architecture. The ablation studies and experimental discussions are also lackluster in this sense, in my opinion: the authors' discussion doesn't go much further than "the results demonstrate method X outperforms method Y", not managing to provide deeper insight into any of the design choices.

The best-motivated addition---the SGRU update---is in my opinion a pretty neat idea and should be more highlighted and motivated in the paper, perhaps with experiments specially designed to show its benefits. Gated attention as well as split messages have already been featured or attempted (in some form) by prior work: see, for example, the GaAN model from Zhang et al. (UAI 2018), hence the novelty of such proposals, in isolation, is limited.

While the GR-GAT model achieves some strong outcomes on synthetic benchmarks, the strength of these results is, in my opinion, insufficient to carry the weight of the paper, especially for a venue like ICLR. Especially considering that these graphs are designed with rather simple edge types (such as edge direction), the value of these results when transferred to real-world heterogeneous graphs is unclear. Further, the max-tree benchmark, where most of the interesting ablations are shown, is a bit concerning: it appears as if it could be quite easy to solve if using max-aggregation rather than attention (see Richter and Wattenhofer's "Normalized Attention Without Probability Cage" for some motivation on this), and maybe in this sense would not require any of the "heavy artillery" proposed here.

As mentioned above, it would be interesting to create more targeted and diverse synthetic benchmarks, perhaps to specifically battle-test the SGRU component.

========= Post-rebuttal update:
I thank the authors for carefully addressing my comments, as well as other reviewers'.

Ultimately, this is a nice paper with a novel recurrent component, and I can see how it could perform well in practice.
However, the lack of stronger real-world experimentation (on datasets such as OGB) unfortunately renders the contribution insufficient -- the synthetic benchmarks being insufficient on their own to pull the weight of the paper.

I retain my score, but encourage the authors to carefully revise and resubmit for the next venue should the paper be rejected.

---

> ### Author Response · Authors · 2020-11-17
> **Response to Reviewer 1**
>
> Thank you for taking the time to review our work and providing helpful comments.
> We are updating our paper using your comments and hope to provide it to you as soon as possible.
>
> See below for detailed responses to the concerns you raised:
>
> --> “My main concern is that the authors do not properly motivate most of their design choices, or properly ground them…” and “The best-motivated addition---the SGRU update---is in my opinion a pretty neat idea and should be more highlighted...”
> We will more clearly justify the issue with backpropagating horizontally that is present in the most popular message passing networks and present the SGRU more clearly as a solution for that problem. We pushed the discussion for GGNN to appendix but we have more space now.
>
> The SGRU was the first change from GGNNs we investigated but found that it alone was insufficient to solve the toy tasks. The other additions were also motivated by improving the backpropagation to neighbour states, which we will motivate better. We will reorganize the explanation of the approach and use the additional space to extend the motivation and discussions.
> We will also run experiments for the first task to test the SGRU and GRU in isolation. In addition, we will try to run additional ablations for the tree max task to see whether the SGRU improves performance compared to other update options. Do you have any other tasks in mind?
> We would also like to remark that GaAN  implements a predictor of head importance, and thus  enables the model to learn to ignore certain heads after aggregation. Compared to this, the (S)GRU node update function enables element-wise control (rather than head-wise). Also, note that GaAN doesn’t address the problem of learning long-range patterns during graph encoding.
>
> --> “While the GR-GAT model achieves some strong outcomes on synthetic benchmarks, the strength of these results is, in my opinion, insufficient to carry the weight of the paper…”
>
> We agree that more experiments on real data will strengthen the paper. To this end, we will experiment with molecular datasets from OGB.
> Regarding the tree max task, in the latest version (which we will provide as soon as possible), we use a more challenging semi-supervised version of the task (which is now supervised at every node).
>
> Thank you for the paper suggestion!
>
> We can try to run some max-aggregated baselines and/or ablations on tree max to verify your concerns. Which settings do you think would be most interesting?

---

> > ### Comment · AnonReviewer1 · 2020-11-18
> > **Follow-up on initial response**
> >
> > Thank you for your response.
> >
> > For now, I will await the updated motivation and explanation for individual model components, and results on OGB datasets.
> > In my opinion, having such results (as well as making sure that max-aggregator architecture doesn't already solve the max-tree task), is critical before the paper is (close to) being ready.
> >
> > Less importantly, I would like to highlight that my GaAN suggestion didn't mean to say that your entire work is not novel.
> > To clarify: I think at least the SGRU component is novel. The GaAN reference is just given as an indication that the idea of gating an attention mechanism, in itself, is not novel.

---

> ### Comment · Area_Chair1 · 2020-11-18
> **Author response**
>
> Dear AnonReviewer1,
>
> We are now entering the second discussion stage. Could you please check whether the authors have addressed your concerns and questions and potentially ask any further clarification questions?
>
> Thank you,
> Your Area Chair

---

### Official Review · AnonReviewer4 · 2020-10-30
**Interesting analysis but writing needs improvement + Need more real world datasets**

**Rating:** 7
**Confidence:** 5

**Review:**


Summary:

    The authors propose a new gating based recurrent graph attention networks for multi-relational graphs to capture long-range neighbor dependencies. The authors provide an interesting analysis of current gated GNN models (in the appendix + Figure 3) in light of their ability to capture long-range dependencies in graphs. Experimental results are reported for node classification with two synthetic datasets and two real-world datasets.

——

Pros:

	(i) The work addresses an important problem of long-range dependencies with conventional graph neural networks. The work has an interesting backpropagation based explanation of the issue in the Appendix and Figure 3 provides a good illustration of the same.
	(ii) The synthetic experiments are interesting and helpful to evaluate models for long-range dependencies
	(iii) Impressive results on synthetic experiments

——
Major Concerns:

	(i) Though the problem of interest is explained well in the appendix from the view of Gated GNNs. The paper's main section lacks a clear explanation of the problem; especially, there is no explanation of what it means by the horizontal vanishing gradient problem.

	(ii) The model motivation is not clearly written in the main paper — why model both the inputs as states of a gated recurrent network. Eqn: 10 is not straightforwardly clear why the redundant combination of information is helpful. Overall, the primary contribution discussed in section 3.3.1 needs to be expanded and explained in contrast to GRU update to capture long-distant neighbor information. A similar backpropagation analysis for the proposed model will help us understand the power of the proposed model. Also, why is the model called 'symmetrically' GRU?

	(iii) The r_x gate in Eqn: 9 is similar to the forget gates in LSTMs. How does it compare with the GraphLSTM updates?

	(iv) Residual and Dense connections are not discussed and experimented. Like in JK-Nets, dense connections can be added to the each of the GCNs pertaining to different relations or can be added for the combined layer output from all relations. The baselines with highway connections need to be evaluated.

	(v) There are too many components or design choices proposed/made, but there are no ablation studies on all the components. (a) Gated relational message, (b)Redundant usage of relational information for attention keys, (c) Concat-Ensemble of value vectors (d) symmetric GRU. Even in the current set of variations, would like to see, GR-GAT(SGRU) - value transformation, GR-GAT(SGRU)


	(vi) Results are reported only for two real-world datasets. MUTAG and BGS can be added. No real-world datasets or tasks with potential long-range dependencies are experimented. Ex: Molecular graphs (MUTAG, ZINC, etc. ) and Protein graphs.

	(vii) Tree Max:
		- A height wise results+analysis of node-level task would be interesting.
		It is especially hard to understand why the GR-GAT(Ident)- value transform performs poorly. On the same note, how does GR-GAT(SGRU) - value transformation perform?
		- Why does GR-GAT(GRU) perform way poorer than GR-GAT(Ident) ?
		- The text about model variations mentions GATE and SGRU to be the same but in Table: 2, there is both GR-GAT (SGRU) and GR-GAT(Gate).

	(viii) WGCN results on AIFB ?

	(ix) Only node classification results reported, in which case the scope of the model and results studied should be explicitly mentioned to be restricted to node classification if that is the intent. If that is not the case, additional link prediction results like RGCN or other tasks should be reported.

		Also, Additional results on single-relational homogeneous graphs can help disentangle the effect of the proposed relational module from the main contribution, the gating mechanism.


Overall Recommendation:

       The paper has interesting content, but the paper is not well organized and motivated well. There is sufficient merit if the analysis could be reformulated and generalized for all message-passing models — the horizontal long-term dependency. The backpropagation based vanishing gradient issue discussed is limited to Gated GNNs alone. It is essential to discuss residual and dense connections too. On the experimental front, adding more real-world datasets would strengthen the paper.
——
Post Rebuttal
Increased the score from 6 to 7.
I would have strongly recommended the paper if it had more real-world datasets.

---

> ### Author Response · Authors · 2020-11-17
> **Response to Reviewer 4**
>
> EDIT: formatting
>
> Thank you for taking the time to review our work and providing helpful comments.
> We are updating our paper based on your comments and hope to provide it to you as soon as possible.
>
> See below detailed responses to the concerns you raised:
>
> (i) We will improve the explanation of the problem and improve the introductory and motivating parts of the paper to more clearly focus and define the issue with backpropagating “horizontally” in GNNs.
>
> (ii) Perhaps the name “symmetrically gated” is not the best here. We use it to refer to the fact that both inputs (which in the case of a GNN corresponds to the aggregated information from the neighbours) and states are treated similarly, and both backpropagation to the neighbourhood aggregation vector as well as previous node states benefits from the same long-range modeling advantages of normal GRUs. With the extra space now, we will extend the discussion in 3.3.1, so that it better explains the contribution and better follows on the improved motivating parts (see also the previous point).
>
> (iii) The proposed method is also motivated by the properties of TreeLSTMs and GraphLSTMs, because in these models the information is propagated horizontally better than in the standard GNNs.
> Regarding Graph LSTMs in general, we found several variants with small differences in the literature. For example, Peng et al. 2017 uses neighbour-specific forget gates. Compared to that work, we use an attention mechanism instead of forget gates, which have to learn independently. We will also discuss this in related work.
>
> (iv). That’s a good suggestion. We will try a residual version of the gated functions in the future.
>
> (v). In table 2, we have some ablations of the components separately, including using \mu_MM instead of the proposed message function. We currently provide GR-GAT(Ident) - value transformation and GR-GAT(SGRU). We can include GR-GAT(SGRU) - value transformation. But because of the many small design choices, it is difficult to provide an extensive ablation due to many possible combinations. That’s why we focused on the presented ones, which we deemed most important.
>
> (vi). We will try to run experiments on the molecular datasets from OGB.
>
> (vii). We ran such an analysis and found that there is no clear picture where more distant nodes get much worse performance than closer nodes. However, perhaps this is not very surprising given that all nodes are equally far from each other in terms of update distance, as we shortly illustrated in the discussion for the first experiment.
> GR-GAT(GRU) vs GR-GAT(Ident): The Ident variant does not have an update function and relies solely on the attention mechanism to build representations. The GRU performs worse because of the “horizontal” backpropagation problem we outlined earlier and elaborated on in more detail in Appendix (the GGNN discussion).
> SGRU and Gate are not the same. The gate variant reuses the same architecture as in 3.1. We will drop the gate variant from the discussion.
>
> (viii) WGCN on AIFB: We could not find WGCN results for AIFB in the literature.
>
> (ix) We focused primarily on node classification experiments because (1) it seems like an easier training setup than link prediction and (2) graph classification relies on a final pooling layer that builds a single representation of the graph. To elaborate on the last point, we think that in graph classification, many patterns spanning a large distance can be captured by the final pooler so if you just take care of the depth-wise skip connection, you could get similar performance as a model that also backpropagates better “horizontally”. In node classification, however, every node has a potentially different representation. So a node classification task seems like a more challenging setup.
> That being said, we will try to include additional experiments on the molecular datasets from OGB, which include graph classification tasks.

---

> > ### Comment · AnonReviewer4 · 2020-11-18
> > **Follow-up**
> >
> > Thank you for your response.
> >
> > - I hope the problem and the proposed solution will be well motivated and explained well in the updated version.
> > - As you also agree that the GraphLSTMs can similarly better model horizontal dependencies, it would be useful to also include it in the experimental analysis.
> > - I strongly recommend to include a backpropagation based analysis for dense+residual connections and GraphLSTMs, similar to GGNNs. This will strengthen the motivation of the paper significantly.
> > - Only two real-world datasets are reported and that too on AIFB the proposed model is compared against only two baselines. At the minimum, I would like to see results for WGCN which was compared on the other dataset.
> > - I partially agree with your motivation to not evaluate the graph classification task but not on the link prediction task. Albeit, it is not a deal-breaker for me provided that the node classification experiments are strong.

---

> ### Comment · Area_Chair1 · 2020-11-18
> **Author response**
>
> Dear AnonReviewer4,
>
> We are now entering the second discussion stage. Could you please check whether the authors have addressed your concerns and questions and potentially ask any further clarification questions?
>
> Thank you,
> Your Area Chair

---

### Author Response · Authors · 2020-11-23
**Rebuttal revision**

Dear readers,

We just updated our manuscript using your feedback. We are looking forward to receive your feedback on this updated version.

The following changes have been done:

Section 3 (motivation) is added to better motivate the particular problem we are tackling.

Section 4 is reordered, shortened, and updated to better motivate the different design choices. In particular, the SGRU has been moved to the beginning (4.1), the description of the attention mechanism (4.2) is shortened (omitted text is in Appendix) and motivated better (first paragraph), and additional motivation is provided for the message function (4.3).

There are some significant changes and additions in experimental section (Section 5) as well:
(i) For the first task (5.1), we verify that the poor performance of GGNN is not due to a deeper network.
(ii) Here, we also augmented the ablation study to include a direct comparison to a model consisting only of a SGRU, compared to a GRU.
(iii) We change the second task/synthetic dataset (5.2) to a more challenging, semi-supervised version and ran an additional setting that illustrates the importance of the SGRU as opposed to the GRU.

Thank you for reading!

---

### Decision · Program_Chairs · 2021-01-07
**Final Decision**

**Decision:**

Reject

**Comment:**

This paper proposes a GNN architecture for multi-relational data to better address long-range dependencies in graphs. The proposed GR-GAT model is a variant of graph attention networks (GAT) with, among other modifications, vector-based edge type embeddings and GRU-type updates. Results are presented on AIFB, AM, and on synthetic benchmarks.

The reviewers agreed that this is an interesting contribution and that the results on the chosen synthetic benchmarks are insightful, but that experimental evaluation on real data and overall motivation of the architecture is lacking. In the rebuttal period, the authors have improved the writing and strengthened the motivation of the paper. However, given the limited amount of time, the authors were not able to sufficiently address the lack of experimental validation on real data (beyond AIFB & AM). I am inclined to agree with the reviewers that this paper needs significantly more work on the experimental evaluation, the overall presentation needs to be refined and it needs to more carefully analyse the effect of each individual architectural modification to meet the bar for acceptance.